Immortalization of American miniature horse-derived fibroblast by cell cycle regulator with normal karyotype

http://orcid.org/0000-0001-5276-8941 Tani Tetsuya ttani@nara.kindai.ac.jp
Department of Advanced Bioscience, Kindai University , Nara, Nara , Japan
Haraguchi Tokuko
Electronic publication date: 2024 Jan 26
Publication date: 2024
Volume: 12
Electronic Location ID: e16832
Received 2023 Apr 6; Accepted 2024 Jan 4
Copyright: © 2024 Tani
Copyright year: 2024
Copyright holder: Tani
License: This is an open access article distributed under the terms of the Creative Commons Attribution License, which permits unrestricted use, distribution, reproduction and adaptation in any medium and for any purpose provided that it is properly attributed. For attribution, the original author(s), title, publication source (PeerJ) and either DOI or URL of the article must be cited.
License URL: https://creativecommons.org/licenses/by/4.0/

Keywords: immortalization, Cell culture, Miniature horse

Funding: Basic budget of Kindai University This research was supported by the basic budget of Kindai University. The funders had no role in study design, data collection and analysis, decision to publish, or preparation of the manuscript.

==============================
Immortalized cells serve as a crucial research tool that capitalizes on their robust proliferative properties for functional investigations of an organism. Establishing an immortalized American miniature horse cell line could yield valuable insights into these animals’ genetic and physiological characteristics and susceptibility to health issues. To date, immortalized small horse cells with normal karyotypes have not been established. In this study, we successfully established primary and immortalized fibroblast cell lines through the combined expression of human-derived mutant cyclin-dependent kinase 4 (CDK4R24C), cyclin D1, and Telomerase Reverse Transcriptase (TERT), although CDK4R24C and cyclin D1, SV40T and TERT did not result in successful immortalization. Our comparison of the properties of these immortalized cells demonstrated that K4DT immortalized cells maintain a normal karyotype. Ultimately, our findings could pave the way for the development of targeted interventions to enhance the health and well-being of American miniature horses.

Introduction

The development of in vitro cell culture systems reduces the need for in vivo experimentation and allows more extensive studies of cell physiology, disease elucidation, and drug testing, cost-effectively and without ethical concerns. Not only immortalized mammalian tissue cell lines have been created as models to mimic pharmacological effects and developmental mechanisms, but also as genetic cell resources, induced pluripotent stem cell (iPSC) and somatic cell cloning.

Equine as livestock has long been established as an international industry of great economic value, as well as being widely used for agriculture. In particular, miniature horses and ponies are officially part of the pony group, are both small equine breeds that are often used as pets, therapy animals, and as a model for the mechanism of body growth in mammals. The very popular American miniature horse was a breed established through crossbreeding between smallest breed Falabella from Argentina and oldest known horse breeds Shetland miniature horse from Shetland island (Heck, Sanchez-Villagra & Stange, 2019). American miniature horses are created based on strong inbreeding and bred to look like scaled-down versions of horses, with a muscular body, a smooth coat, a long neck, straight legs, and a short back. They are usually less than 34 inches tall at the withers (Brooks et al., 2010). Therefore, new technologies or experimental evaluations that can further study the physiology of small horses to improve their health and well-being will have a strong economic impact. Therefore, we felt that the development of an in vitro immortalized American miniature horse cell line would have great potential for understanding equine skeletal muscle metabolism, combating infectious diseases, and evaluating the rapid evolution of size during domestication.

Immortalized cells have been proven in numerous cells to escape cellular senescence and overcome cell growth limitations. Many types of mammalian cells have been established by gene transfer of the simian virus large T antigen (SV40T) or human papilloma-virus-derived E6E7 protein (HPV-E6E7), which binds and inactivates not only the cell cycle-inducing retinoblastoma protein (pRB) (Ahuja, Saenz-Robles & Pipas, 2005). However, because the p53 protein is inactive, the production of SV40 or HPV-E6E7 commonly results in chromosomal abnormalities (Schmidt-Kastner et al., 1996). Therefore, virus-derived immortalized cells often have different properties from parental cells (Takada et al., 2021). In addition, immortalized cell lines derived from human TERT are commonly used in many animal species while maintaining normal chromosomes (Counter et al., 1998).

To date, our research group has successfully established mammalian immortalized cells in many species that retain the same cellular characteristics as the original cells by utilizing the expression of the cell cycle regulators mutant human-derived cyclin-dependent kinase (CDK4R24C), cyclin D1, telomerase reverse transcriptase (TERT) (Donai et al., 2014; Fukuda et al., 2016; Orimoto et al., 2020b; Fukuda et al., 2018; Katayama et al., 2019; Orimoto et al., 2020a; Tani et al., 2019). However, the optimal immortalization method in American miniature hoses has not been investigated to date.

This study aims to establish an immortalized American miniature horse cell line by introducing CDK4R24C and cyclin D1 (name as K4D), K4D plus TERT (K4DT), TERT and SV40T. The establishment of an immortalized cell line would allow for the production of a stable and continuous source of cells for both genomic and functional studies. The immortalized cell line would also provide an opportunity to investigate the molecular mechanisms underlying the susceptibility of American miniature horses to health concerns and potentially develop therapeutic interventions to mitigate their impact, as well as explore potential applications in iPSC (Ezashi, Yuan & Roberts, 2016) and somatic cell cloning (Galli et al., 2008).

Materials and Methods

Unless otherwise noted, all chemicals and media were obtained from FUJIFILM Wako Pure Chemical Corporation (Osaka, Japan).

The primary cell of American miniature horse

A tissue sample was obtained from an American miniature horse that had died of natural causes in a private Zoo (World Ranch, Osaka, Japan). The 15-year-old male and 13-year-old male individuals were registered by The American Miniature Horse Association as a pet. Muscle fragments were obtained from a syringe injected into the hind leg muscles and primary cells were established by explant culture in high glucose DMEM supplemented with 10% fetal bovine serum (FBS; Gibco, Carlsbad, CA, USA) and 1% penicillin-streptomycin-amphotericin mixture and 1% Zellshield (Minerva Biolabs, Berlin, Germany) at 37 °C in a CO2 incubator for 10 days. After several passages by 0.25% trypsin-1 mM EDTA solution, the primary cells were suspended in CultureSure cell freezing medium in a liquid nitrogen tank until use. To confirm mycoplasma contamination, established cells were subjected to PCR Mycoplasma Detection kit (Takara Bio, Shiga, Japan).

Preparation of recombinant viruses for gene transfer into primary cells

To prepare the lentivirus for gene transfer into primary cells, we obtained all relevant plasmids from Dr. Hiroyuki Miyoshi (RIKEN BioResource Center, Tsukuba, Japan) or Addgene (Watertown, MA, USA). We used the expression plasmids CSII-CMV-CDK4R24C-P2A-Cyclin D1-T2A-EGFP, pLV-CMV-TERT-PGK-Bsd (Vectorbuilder, Chicago, IL, USA), and pLV-CMV-SV40T-PGK-Puro (Vectorbuilder). We introduced the packaging plasmids psPax2 (Addgene #12260) and pMD2.G (Addgene#12259) into 293T cells (RCB5708; RIKEN BioResource Center, Kyoto, Japan) using lipofection. The virus supernatant was recovered after 48 h transfection and filtered using a 0.45 mm disk filter (Merck Millipore, Burlington, MA, USA) in accordance with the Addgene lentivirus protocol (https://www.Addgene.org/protocols/lentivirus-production/). The primary cells were incubated with the lentivirus at a multiplicity of infection (MOI) of five for each virus, supplemented with 8 μg/ml of polybrene (Sigma) for 24 h. To estimate the efficiency of virus infection, enhanced green fluorescent protein-expressing cells (K4D-EGFP) were used as a control. Infected cells of TERT and SV40T lentivirus were selected using culture containing 500 ug/ml hygromycin or 10ug/ml blasticidin for 5–7 days.

Population doubulings and cell doubling time assay

Primary and established cells were seeded in 24-well plates at a density of 1.0 × 10^4 cells/well, the cells were cultured when one well of the six-well plate became confluent. Cell passages were repeated until they could no longer continue. The population doubling level (PDL) was calculated using the formula log2 (A/B), where A is the number of cells harvested in a passage and B is the number of cells seeded. The results of the PDL assay were plotted as the average of three consecutive samples. Cell doubling time were calculated using the formula T × (ln2)]/[ln (Xe/Xb)], where T is the incubation time, Xb is the number of cells at the beginning of the time incubation, Xe is the number of cells at the end of the incubation time, and ln is the Napierian logarithm.

BrdU incorporation assay

To detect the cell ratio of proliferating cells based on the measurement of 5-bromo-2′-deoxyuridine (BrdU) incorporation during DNA synthesis, 10 μM of BrdU (Cayman, Ann Arbor, MI, USA) incorporated for 3 h at 37 °C in a CO2 incubator. After several wash in PBS, fix and permeabilized by 4% PFA and PBS containing 0.1% Triton-X-100 (PBT). Then, cells were incubated overnight with a rabbit polyclonal anti-Brdu antibody (1:1,000; Bioss Antibodies Inc, Woburn, MA, USA) in PBS-1%BSA at 4 °C. After several washes in PBS containing 0.1% Tween20, they were incubated for 60 min at room temperature with a secondary antibody: an Alexa-Fluor-594-conjugated goat anti-rabbit IgG antibody (1:1,000; Thermo Fisher Scientific, Waltham, MA, USA). The cells were washed several times in PBS, mounted on a glass slide with the ProLong Gold Antifade Reagent and 10 μg/ml Hoechst 33342 solution, and were examined under a fluorescence microscope EVOS M5000 (Thermo Fisher Scientific, Waltham, MA, USA). BrdU incorporation was quantified by counting at least 5,000 nuclei in each established cell.

Senescence-associated β-galactosidase staining assay (SA- β-gal)

To detect cellular senescence, SA-β-Gal staining was carried out at passage 15, with Cellular Senescence Detection Kit (CELL BIOLABS, San Diego, CA, USA). Cells were fixed at RT for 10 min, washed with PBS, and stained in β-galactosidase staining solution containing X-gal (pH 6) at 37 °C overnight in dry incubator. Stained cells were monitored under bright field microscopy.

DNA damage-associated γH2AX staining

To detect the DNA damage, the cells were immune-stained with a monoclonal anti-γ H2AX antibody (1:1,000; Abcam, Cambridge, UK). Counting of gamma H2AX foci in a cell was quantified by counting at least 50 nuclei in each established cell by a confocal microscopy FV-3000 (Olympus, Tokyo, Japan).

ATP measurement

Established cells were washed several times with PBS. The intracellular ATP content in 1.0 × 104 cells was quantified using Cell Titer-Glo Luminescent cell viability assay (Promega, Tokyo, Japan).

Immunostaining

To identify the type of cells established, primary cells were immunostained with anti-vimentin, cytokeratin and desmin antibodies. Immunofluorescence staining of the cells on coverslips was performed as BrdU incorporation assay using anti-vimentin (1:1,000; Bioss Antibodies Inc, Woburn, MA, USA), anti-cytokeratin (1:1,000; Bioss Antibodies Inc, Woburn, MA, USA) and anti-desmin (1:1,000; Bioss Antibodies Inc, Woburn, MA, USA). As a negative control, all cells were incubated with nonimmune rabbit serum instead of rabbit polyclonal primary antibodies.

Telomerase repeated amplification qPCR (TRAP-qPCR) assay

Telomerase activity was measured by TRAP-qPCR as previously with slight modification (Pinto et al., 2021). In brief, quantitative Real-Time PCR was performed using Thunderbird SYBR qPCR Mix (Toyobo, Osaka, Japan) in a final reaction volume of 10 μl. The 293T cells standard curve were used as Relative Telomerase Activity (RTA) quantification.

F-actin staining

To detect of cell morphology, cytoskeleton, cell motility, and polarity, the cells were stained with a rhodamine X conjugated-phalloidin and Hoechst 33342. In brief, fix and permeabilized by 4% PFA and PBT were incubated for 60 min at room temperature with 0.1% of rhodamine X conjugated-phalloidin solution.

Chromosome analysis

To determine the chromosomal normality, the chromosomes of the established cells were counted using the Giemsa staining method. In brief, cells were treated with 50 ng/ml demecolcine solution for 2 h to arrest the cell cycle at metaphase, after trypsinization, then we treated the cells with a hypotonic solution of 0.075 M potassium chloride and fixed using Carnoy’s fixative (75% methanol/25% acetic acid). Post-fixation, the cells were stained with Giemsa solution. We analyzed the detailed chromosomal patterns of 20 mitotic cells.

Detection of inserted gene cassettes by genomic polymerase chain reaction (PCR)

Genomic DNAs was extracted using NaOH-base extraction method. In brief, 1.0 × 104 cells were heated in 90 μl of 50 mM NaOH solution at 100 °C for 10 min and neutralized with 10 μl of 1 M Tris-HCl solution at pH 7.5. Genomic DNAs were subjected to genomic PCR by KOD one enzyme (Toyobo, Osaka, Japan) according to manufacture (2 min of pre denaturation at 94 °C, 40 cycles of 10 s at 98 °C and 20 s at 68 °C). For detection of introduced gene cassettes as well as 12s rRNA gene as a positive control. The sequences of the CDK4R24C-2A-cyclin D1 primers were

5′-GCTGGAGATGCTCACCTTCA-3′ and 5′-TCCAGGTGGCCACGATCTTTC-3′, TERT 5′-ACTACCGCGAGGTGCTG-3′ and 5′-ACCGTGTTGGGCAGGTA-3′, SV40T 5′-TTCTTACTCCACACAGGCATAG-3′ and 5′-ACTCCAGCCATCCATTCTTC-3′, 12srRNA 5′-CAAACTGGGATTAGATACCC-3′ and 5′-CAGGAAACAGCTATGACC-3′. The PCR products were detected the in 1.8% agarose gel electrophoresis with ethidium bromide.

Identification of the species of animal

To confirm animal species for avoid cell contamination, mitochondrial 12S ribosomal RNA gene was amplified by PCR and sequenced as previously reported (Yang et al., 2014). In brief, isolation of genomic DNA were isolated from cultured cells by Template Prepper for DNA kit (Nippongene, Tokyo, Japan). A total of 10–100 ng of DNA were used for PCR amplification and 12S rRNA gene was amplified the following primers;

5′-TGTAAAACGACGGCCAGT CAAACTGGGATTAGATACCC-3′ and

5′-CAGGAAACAGCTATGACCGAGGGTGACGGGCGGTGTGT-3′ in KOD One PCR enzyme according to manufacture. The PCR product was direct sequenced by M13 forward or reverse primer and analyzed via Standard Nucleotide BLAST searches at the NCBI website.

Statistical analysis

Statistical analyses were performed using GraphPad Prism 10 (GraphPad Software, La Jolla, CA, USA). All the data shown are the mean ± SEM from at least six biological replicates. Significance was assessed compared with the primary cells as controls using One-way ANOVA with post-hoc Dunnett’s test. A p-value less than 0.05 was considered statistically significant.

Results

Cell culture and morphology of established cell lines derived from American miniature horses

Primary cells were isolated from muscle tissue fragments obtained via syringe biopsy from two male American miniature horses (Fig. 1A). Under standard culture conditions, the primary cells exhibited a fibroblast-like morphology (Figs. 1B and 1C). To confirm the cell type of the primary cells, primary antibodies against vimentin were used to stain the cells. As shown in Fig. 1D, the primary cells were found to be vimentin positive. To immortalize American miniature horse fibroblasts, primary cells were transduced with a lentivirus composed of K4D-EGFP with or without TERT and SV40T. The efficiency of transduction was monitored via the expression of EGFP, which was confirmed 5 days after transduction. The primary American miniature horse fibroblasts were efficiently infected with K4D-EGFP-expressing virus, resulting in a high percentage of green fluorescence-positive cells (Fig. 2A). The expression of TERT and SV40T was selected by multiple passages in the presence of blasticidin or puromycin.

Figure 1 Establishment of fibroblast from American miniature horse muscle fragments.

(A) Muscle fragments sample. (B) Fibroblasts-like cells spread from muscle explant. (C) A typical spindleshaped fibroblast in passage 5. (D) Immunostaining showed green for vimentin and blue for nuclei. Scale bars = 100 um.

Figure 2 Detection of K4D-EGFP expression and genomic integration of introduced genes cassette in American miniature horse-derived cell.

(A) Five days after infection, EGFP expression via CS-CMV-K4D-EGFP in American miniature horse-derived cells was detected, whereas no infected wild type was the negative control (miniature horse #1). Scale bars = 50 um. (B) Detection of genomic integration of expression cassette by polymerase chain reaction (PCR). PCR amplification with expression cassette for CDK4-Cyclin D1, TERT, SV40T and internal control mitochondrial gene 12s-rRNA.

The established cells by K4D, K4D plus TERT, and SV40T were named as K4D, K4DT and SV40T cells respectively. To detect introduced gene cassette, I performed genomic PCR using the genomic DNA obtained from the primary, K4D, K4DT, SV40T and TERT cells derived from two individuals. As shown in Fig. 2B, expected size of CDK4-2A-CyclinD1 fragments in K4D and K4DT cells, TERT fragments in K4DT cells, and SV40T fragments in SV40T cells were detected. These results indicate that the target genes were successfully introduced into the transduced cells.

K4D, K4DT and SV40T immortalized cells of American miniature horse-derived cells are free from cellular senescence

As shown in Fig. 3A, sequential passages of primary, K4D, K4DT, SV40T, and TERT transduced cells derived from two individuals were carried out for 6–7 months. Although the primary and TERT cells of two individuals could not continue cell proliferation beyond 15 passages (25 PD value). Both K4D and SV40T cells of stop cell proliferation around 65 PD and 80 PD.

Figure 3 Biological analysis of proliferation in American miniature horse-derived cells.

(A) Cell growth curve under the sequential passages. (B) Doubling time at passage 10. (C) BrdU incorporation for 3 h at passage 10 was detected by BrdU antibody. (D) Detectionof SA-β-Gal positive cells at passage 15. Independent biological experiments, n = 6. The data are shown as mean ± SEM. One-way ANOVA with Dunnett’s test p-values is indicated as *p < 0.05, **p < 0.001 compared to the control.

While both K4DT cells showed cell proliferation of more than 100 PD values. The doubling time at passages 10 were 32.1 h (miniature horse #1) and 34.4h (miniature horse #2) for primary when compared to K4D, K4DT, and SV40T cells significantly (Fig. 3B). Additionally, the BrdU incorporation rate for both primary cells were significantly lower when compared to K4D, K4DT, SV40T and TERT cells (Fig. 3C). To assess the degree of cellular senescence, SA-β-gal staining was performed at passage 15. Both primary and TERT cells almost stopped proliferating and stained positive for SA-β-gal. However, K4D, K4DT, and SV40T cells did not stain for SA-β-gal (Fig. 3D). These data showed that K4D, K4DT, and SV40T expressing cells from two individuals dramatically prolonged the proliferation up to 65 PD values of American miniature horse-derived cells. While both K4DT cells did not stop proliferation over 100 PD.

Biological characterization of established American miniature horse-derived cells

For the purpose of comparing the biological characteristics of the established cells from two individuals, I conducted an analysis of ATP content, γ-H2AX immunostaining, and telomerase activity for miniature horse#1. The ATP contents of the established cells were found to be almost equivalent (Fig. 4A). Figures 4B and 4C displayed that SV40T cells had a higher number of γ-H2AX foci in a cell significantly when compared to primary, K4D, K4DT and TERT cells, which resulted in a greater accumulation of DNA damage under these culture conditions. Finally, I compared the telomerase activity, and the results presented in Fig. 4D demonstrated that K4DT and TERT cells exhibited enzymatic activity for extending the telomere sequence through TERT enzyme activity significantly.

Figure 4 Biological characteristics of American miniature horse-derived cells.

(A) ATP contents were measured luminescence. (B and C) Immunostained by anti-gamma-H2AX antibody and counted foci numbers in a cell (D) Telomerase activity by TRAP-qPCR assay as relative telomerase activity to 293T cell extract (RTA). Scale bar = 20 um. Independent experiments, n = 6. Independent biological experiments, n = 6. The data are shown as mean ± SEM. One-way ANOVA with Dunnett’s test p-values is indicated as *p < 0.05, **p < 0.001 compared to the control.

Karyotype and F-actin distribution of primary and K4D American miniature horse-derived cells

To further investigate F-actin distribution and the karyotype of primary and K4DT American miniature horse #1-derived cells were analyzed. Utilizing phalloidin and Hoechst33342 labeling, I examined the distribution of F-actin in the primary, K4D, and K4DT cells. Figure 5A displayed that none of the cell types’ F-actin distribution or structure. Next, I analyzed twenty metaphase plates for primary (passage 5 and 6), K4D (passage 16 and 11), and SV40T (passage 21 and 24) from two individuals. As illustrated in Figs. 5B and 5C, over 70% of primary, K4D, K4DT and TERT cells exhibited normal karyotypes (2n = 64) of both individuals. However, all SV40T cells of both individuals had abnormal chromosome numbers ranging from 69 to 88, with an average chromosome number of 77.33 and 78.1, respectively. These findings demonstrate that K4D, K4DT and TERT cells have a similar chromosome number and karyotype as the primary cells, whereas SV40T cells differ from the primary cells.

Figure 5 Cytoskeletal F-actin distribution and Karyotyping of American miniature horse-derived cells.

(A) F-actin and nuclei were stained in rhodamine X-conjugated-phalloidin (red) and Hoechst 33342 (blue) on primary and K4DT cells (miniature horse #1). Scale bar = 100 um. (B) Chromosome spread of K4DT cells. Scale bar = 10 um (C) Chromosome number distribution on primary and transduced cells. Twenty mitotic cells were analyzed. The data are shown as mean ± SEM. One-way ANOVA with Dunnett’s test p-values is indicated as **p < 0.001 compared to the control.

Identification of the species of origin by mitochondrial 12s rRNA

The amplified fragment of mitochondrial 12S rRNA from K4DT cells derived from two individuals was sequenced using universal oligodeoxynucleotide primers with M13 sequence, and the resulting sequence was determined by searching the NCBI database, as previously reported (Yang et al., 2014). The sequence demonstrated complete homology with the 12S rRNA genes in Equus caballus (MN187575.1; Figs. 1S and 2S). This revealed that both cell lines derived from two individuals had no cross-contamination with other species. Furthermore, this study demonstrated that the established cells were free of mycoplasma contamination using commercial mycoplasma detection kit, although the corresponding data was not presented.

Discussion

This is the first report on the successful culture of fibroblasts and the establishment of immortalized cells from an American miniature horse. In a previous study, we demonstrated that co-expression of mutant CDK4R24C, cyclin D1, and TERT can induce cellular immortalization while maintaining a normal karyotype in various mammalian cells (Donai et al., 2014; Fukuda et al., 2016; Orimoto et al., 2020a, 2020b), such as avian (Katayama et al., 2019) and reptiles (Fukuda et al., 2018). In this study, we assessed the suitability of the CDK4R24C-cyclin D1 complex, with or without TERT, to establish immortalized cells from American miniature horse-derived fibroblasts while retaining the original chromosomal characteristics. In this study, there was no difference in PDL and doubling time, even in the absence of TERT, as previously reported (Gouko et al., 2018). The passaging maximum culture period was 4 months for this experiment; however, continued culture for an even longer period may shorten the telomere length of K4D cells and stop their proliferation, as previously reported (Munirah et al., 2022). Our results indicate that co-expression of the CDK4R24C-cyclin D1 complex with or without TERT is more effective in establishing immortalized cells with chromosomal stability than the commonly used SV40T. Furthermore, we were unable to establish immortalized cells through TERT expression despite its success in many other mammalian cells (Xiang et al., 2000; Buser et al., 2006; Oh et al., 2007; Techangamsuwan et al., 2009). This result agrees with the results of a previous study in which donkeys, Burchelli’s zebra, and Grevy’s zebra were not immortalized by TERT alone (Vidale et al., 2012). The low compatibility of the telomerase subunit in horses with humans is thought to be the reason for this failure, and the authors have succeeded in immortalizing equine fibroblasts by simultaneously expressing TERC and TERT. Nonetheless., the immortalized cells of the donkey and Grevy’s zebra showed tetraploidy. These results indicated that immortalization by telomerase has limited success in eques.

We also observed a higher incidence of DNA damage in SV40T cells, as demonstrated by the presence of γ-H2AX foci and abnormal chromosome appearance. These findings align with those of previous studies that demonstrated the genotoxic effects of SV40T expression. activates DNA damage response pathways and alters the expression of DNA repair genes (Onwubiko et al., 2020). Moreover, SV40T expression has been linked to ROS accumulation of reactive oxygen species, which can result in oxidative damage to DNA and other cellular components (Kapplusch et al., 2022). Thus, our study underscores the importance of choosing appropriate methods of immortalization that do not compromise the genomic integrity of cells.

Traditionally, expression of oncogenic proteins, such as SV40T antigen (Tevethia, 1984), HPV-E6/E7 (Hawley-Nelson et al., 1989) and human telomerase catalytic subunit TERT (Counter et al., 1998), has been employed as an immortalization technique. However, these proteins inactivate the p53 tumor suppressor protein, which is crucial for maintaining genomic accuracy. Consequently, these methods frequently result in chromosomal abnormalities and genomic instability. Additionally, SV40T triggers an IFN-mediated immune response in human cells and induces an antiviral state (Forero et al., 2014). Nevertheless, inappropriate methods of immortalization can lead to genetic instability, heterogeneity, and altered cell phenotypes, all of which can compromise the reliability and reproducibility of experimental results. Recently, our research group compared the transcriptome of immortalized cells generated from the same cell sources using three different methods: SV40, HPV-E6/E7, and K4DT, in resulting K4DT immortalized cell resembled to original cells (Fukuda et al., 2021). This study also demonstrated that the co-expression of K4DT is a more suitable method for establishing immortalized American miniature horse fibroblasts that preserve the original chromosomal characteristics of the cells. This discovery is critical for developing more efficient and reliable methods for cell line establishment, which can enhance the validity and be appropriate for homogeneity (Kato et al., 1998; Yin et al., 2002) and heterogeneity (Tani et al., 2019) somatic cell cloning, establishment of iPSC, and direct reprogramming experiments.

Our group has previously reported that vole K4DT cells can be induced into iPSC by forced expression of six reprogramming factors as primary cells in voles (Katayama et al., 2017). Additionally, human fibroblasts immortalized by TERT can be transdifferentiated into functional skeletal muscle constructs by overexpressing MyoD (Xu, Siehr & Shen, 2020). However, SV40T-immortalized cells disrupt the developmental potential of pig somatic cell cloning (Eun et al., 2017). These findings demonstrate that immortalized cells with normal karyotypes are suitable for pluripotent reprogramming and direct reprogramming experiments. Donor cells for these experiments require a large number of cells because of their low efficiency. However, immortalized cells with normal karyotypes have no supply limit for many experiments.

However, our study had certain limitations. We tested only three different methods of immortalization and did not explore other factors that may affect the chromosomal stability of the cells, such as culture conditions and donor variability. Moreover, we did not conduct functional assays to compare the phenotypes and behaviors of immortalized cells obtained using different methods.

Our study enabled the successful establishment of immortalized cells from American miniature horses. The establishment of immortalized cell lines is crucial for biomedical investigation, pharmaceutical innovation, and clinical practice. Our research contributes to the advancement of more efficient and dependable techniques for establishing cell lines that can augment the soundness and translatability of biomedical research. Moreover, the successful immortalization of fibroblasts obtained from American miniature horses may have significant ramifications for the preservation and supervision of this species.

Conclusions

To our knowledge, this study is the first report on a successful establishment of immortalized cells from an American miniature horse-derived fibroblast using the co-expression of CDK4R24C-cyclin D1 complex with TERT. This technique was more effective in establishing immortalized cells with chromosomal stability than the commonly used SV40T and human TERT induction. The study is critical for developing more efficient and reliable methods for cell line establishment, which can enhance the validity and be appropriate for somatic cell cloning, establishing iPSC, and direct reprogramming experiments.

Supplemental Information

Supplemental Information 1 BLAST result profiles using the PCR amplicon of the American miniature horse mitochondrial 12sRNA gene.

Although the identity is only 100%, the profiles show that the complete PCR amplicon of the American miniature horse’s mitochondrial 12S rRNA gene only matches the Equus caballus (MN187575.1) mtDNA.

Click here for additional data file.

Supplemental Information 2 Sequence of American miniature horse 12srRNA.

Sequence was read by M13 Foward primer.

Click here for additional data file.

Supplemental Information 3 Detection of senescence-associated beta-galactosidase (SA-β-Gal).

Each of the cells was stained by SA-beta Gal stain. Scale bars = 50 um.

Click here for additional data file.

Supplemental Information 4 Raw data of doubling time assay.

Doubling time assay of primary, K4D, K4DT, SV40T and TERT cells derived from miniature horse#1 and #2.

Click here for additional data file.

Supplemental Information 5 Raw data of BrdU incorporation assay.

BrdU incorporation assay of primary, K4D, K4DT, SV40T and TERT cells derived from miniature horse#1 and #2.

Click here for additional data file.

Supplemental Information 6 Raw data of SA-β-Gal assay.

SA-Gal assay of primary, K4D, K4DT, SV40T and TERT cells derived from miniature horse#1 and #2.

Click here for additional data file.

Supplemental Information 7 Raw data of intracellular ATP assay.

Intracellular ATP assay of primary, K4D, K4DT, SV40T and TERT cells derived from miniature horse#1 and #2.

Click here for additional data file.

Supplemental Information 8 Raw data of gamma-H2AX foci counting.

Number of gamma-H2AX foci/cell of primary, K4D, K4DT, SV40T and TERT cells derived from miniature horse #1 and #2.

Click here for additional data file.

Supplemental Information 9 Raw data of Telomerase Repeated Amplification qPCR (TRAP-qPCR) assay.

Telomerase activity assay of primary, K4D, K4DT, SV40T and TERT cells derived from miniature horse#1 and #2.

Click here for additional data file.

Supplemental Information 10 Raw data of the frequency of chromosome number.

The frequency of chromosome number of K4D, K4DT, SV40T and TERT cells derived from miniature horse#1 and #2.

Click here for additional data file.

The author thanks veterinarian Keiji Maeda (Ishii Veterinary Hospital) for sample collection, Dr. Tomokazu Fukuda (Iwate University) and Toru Kiyono (Exploratory Oncology Research & Clinical Trial Center, National Cancer Center) for providing the K4D-EGFP plasmids.

Additional Information and Declarations

Competing Interests

Author Contributions

Data Availability

The authors declare that they have no competing interests.

Tetsuya Tani conceived and designed the experiments, performed the experiments, analyzed the data, prepared figures and/or tables, authored or reviewed drafts of the article, and approved the final draft.

The following information was supplied regarding data availability:

The BLAST result profiles using the PCR amplicons of the American miniature horse mitochondrial 12S rRNA gene are available in the Supplemental File.

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
