# Peer review of "Immortalization of American miniature horse-derived fibroblast by cell cycle regulator with normal karyotype"

_PeerJ, doi:10.7717/peerj.16832_

## Round 0.1 · original submission · Major Revisions

Two reviewers recognize the value of this paper but also point out the need for revision. I agree with them. I hope their comments will help you improve your paper. It seems that reviewer 1's comment has appeared many times, but please ignore it after the second time.

Reviewer 1 ·

Basic reporting

.

Experimental design

.

Validity of the findings

.

Additional comments

Dear Editors and Editor in Chief of Peer J
We carefully reviewed the manuscript entitled "Immortalization of American miniature horse-derived ù‚broblast by cell cycle regulator with normal karyotype" by Tani et al. The author showed the establishment of Amarican miniature horse-derived immortalized cell using the expression of mutant CDK4, Cyclin D1, TERT. Although the presented data is easy to understand, manuscript need to be revised extensively based on the comments in below.

1)How many animals has used for this study?
According to the description in the materials and method, it looks the author showed the results from one individual. To evaluate the reproducibility of the result, at least two animals need to be evaluated. The results from other animals are essential.

2) No evidence of introduced geens
The authors do not show the evidence or data to detect the introduced gene. The authors should need to show the PCR data or western blot of introduced gene products. There is no evidence that support the successful introduction of gene delivery.

3) Statistics
Although the authors described "GraphPad Prism 9.01" in the section of statistics. However, the authors probably do not understand the basics of statistics. In case of more than group 3, use of t-test is big mistake to repeat the t-test. Furthermore, number of replications should be involved in the figure legends. The replication number should be more than 6 at least. In addition, the evaluation of normal distribution is essential to use the t-test.

4) SD or SE?
Although the authors use the SD ion the data presentation, we are wondering if the authors really want to show the data distribution. The author looks that they would like to show the difference of average. In that case, they should use SE, instead of SD.

5)TERT
Although the authors showed that both of K4D and K4DT cells showed good proliferation, they are biologically different from view point of TERT activity. They should show the western blot of TERT.

6)SA-beta staining
Although the author presented staining results of SA_beta, they should involve the raw data of the staining. The staining is quite semi-quantitative. The involvement of all data as supplemental materials are essential to support the solidness of the data.

Reviewer 2 ·

Basic reporting

In this manuscript, the author tried to establish the American miniature horse immortalized cells. I fully agree that the successful immortalization of fibroblasts obtained from American miniature horses may have significant ramifications for preserving and supervising this species. However, there are several concerns before publishing to the journal.

Experimental design

1. The function of TERT in American miniature horse K4DT cells for immortalization
The function of TERT in American miniature horse K4DT cells for immortalization is unclear in this study. Although the author recognized the telomerase activity in K4DT cells (Fig4.D), cell growth is similar between K4D and K4DT cells (Fig.3A). The author should continue the sequential passage or show other reasonable data.

2. Criteria of immortalized cell
The author addressed the sequential passage until around PD55. In general, the criteria for immortalization is PD100. Therefore, I recommend the continue the sequential passage or correcting the main text.

3. Karyotype analysis of K4DT
The Author addresses the karyotype analysis of K4D and SV40T cells. The TERT gene sometimes induces chromosomal instability. Therefore I recommend that the author address the karyotype analysis of K4DT cells, if the K4D cell did not satisfy the criteria of immortalized cells.

Validity of the findings

Nothing.

---

## Round 0.2 · accepted · Accept

Thank you very much for submitting a revised version of your paper. Two reviewers agreed that your paper addresses all of the reviewers' comments. Therefore, I will accept your paper for publication in PeerJ.

Reviewer 1 ·

Basic reporting

Dear Editor in Chief
Now, we confirmed that the author made appropriate modifications of the manuscript.

Experimental design

Now, we confirmed that the author made appropriate modifications of the manuscript.

Validity of the findings

Now, we confirmed that the author made appropriate modifications of the manuscript.

Additional comments

Now, we confirmed that the author made appropriate modifications of the manuscript.

Reviewer 2 ·

Basic reporting

Author fully addressed my comments in this round.
I recommend the acceptance.

Experimental design

Author fully addressed my comments in this round.
I recommend the acceptance.

Validity of the findings

Author fully addressed my comments in this round.
I recommend the acceptance.